# LLM-Assisted Reinforcement Learning for Distributed Scheduling

## Abstract

The distributed flexible job-shop scheduling problem (DFJSP) involves coordinating job execution across distributed factories to achieve production goals. While existing reinforcement learning (RL)-based scheduling methods have shown promise in learning adaptive scheduling polices, they often rely on shallow networks and simple handcrafted rewards. These designs limit global state reasoning and accurate credit assignment under sparse rewards, thereby hindering the ultimately balanced workload distribution and efficient policy learning. To address these limitations, we propose a Large Language Model (LLM)-assisted RL algorithm tailored for DFJSP by leveraging the contextual reasoning and prior knowledge of LLM. Specifically, we propose an LLM-driven factory assignment mechanism that encodes global factory states and job features into structured queries, enabling context-aware and effective coordination among factories. Furthermore, we design an LLM-informed reward model that encodes scheduling-aware semantics into multi-dimensional proxy rewards for precise credit assignment during training. Theoretically, we provide a bound on the reward approximation error and prove that the proposed assignment strategy effectively reduces global workload variance. Extensive experiments conducted on public benchmarks (i.e., Hurink and Brandimarte) and multiple simulated DFJSP instances of varying scales demonstrate that our algorithm consistently outperforms RL-based scheduling methods, achieving the average makespan improvement ranging from 0.61% up to 25.78%. Our source code is available at `https://anonymous.4open.science/r/LaRL-407B`.

## 1 Introduction

Scheduling is a fundamental process that involves managing, coordinating, and optimizing the execution of jobs and workload in a manufacturing system (Li et al., 2024). Among various scheduling problems, the distributed flexible job-shop scheduling problem (DFJSP) has attracted significant attention because it supports geographically distributed production, aligning with modern trends (Huang et al., 2024a). In DFJSP, a set of jobs is assigned to multiple factories to optimize the desired objective (such as makespan or tardiness), after which each job is processed on a group of machines within the assigned factory based on the predefined operation sequence (Zhang et al., 2024). In practice, DFJSP is often subject to unexpected disturbances, especially in custom manufacturing companies, where flexible order placement leads to frequent new job arrivals (Huang et al., 2024b). Therefore, it is critical for these companies to develop effective scheduling algorithms to ensure production efficiency in uncertain environments.

Existing scheduling algorithms can generally be divided into three categories: metaheuristics, heuristics, and reinforcement learning (RL)-based. Metaheuristics (Wang et al., 2025a) explore approximate solutions by iteratively evolving a population of solutions. Heuristics (Ito et al., 2022) obtain feasible solutions by assigning priorities to jobs and factories based on specific criteria. Although the above algorithms can provide reasonable solutions, they typically rely on handcrafted rules and struggle to generalize across different scenarios. Recently, reinforcement learning (RL) has shown strong potential in eliminating handcrafted heuristics and enhancing adaptability in diverse scheduling scenarios (Zhang et al., 2020). RL-based scheduling algorithms can automatically learn a scheduling policy from experience to optimize long-term performance (Lei et al., 2023). One popular line is RL-based heuristic selection (Lei et al., 2024), which leverages RL to select among

predefined heuristics based on states. Another line adopts end-to-end learning (Huang et al., 2024a), where the policy is directly learned from raw features without relying on handcrafted heuristics.

While these RL-based algorithms demonstrate promising results, they still face critical challenges in global coordination and reward design for solving DFJSP. First, **how to enable global coordination under limited state reasoning?** Most RL-based scheduling algorithms assume that factory assignment is handled by shallow policy networks (Huang et al., 2024b) or simple heuristics (e.g., earliest available time and minimum transfer time) (Lei et al., 2024). These approaches lack sufficient state reasoning capacity to capture the global relationships between factories, resulting in poor coordination and load imbalance. Second, **how to achieve effective credit assignment under spare reward?** Existing algorithms often manually design scalar rewards based on the scheduling objectives (Hameed & Schwung, 2023; Lei et al., 2024). However, since these objectives typically depend on a complete scheduling cycle, most intermediate actions receive no meaningful feedback during execution. This leads to ambiguous credit assignment, making it difficult for the agent to distinguish beneficial actions and hindering overall learning efficiency (Qu et al., 2025).

LLMs have shown strong capabilities in capturing global context and reasoning over structured inputs (Achiam et al., 2023), making them well-suited for effectively addressing the challenges of coordination and credit assignment in complex scheduling tasks. Motivated by these, we propose an LLM-assisted RL algorithm, *LaRL*, to facilitate global coordination among factories and adaptive reward design for solving the DFJSP with new job arrivals. Our main contributions can be summarized as follows: 1) We propose an LLM-driven factory assignment mechanism that leverages the contextual reasoning of LLMs to dynamically allocate new jobs based on global factory workload and job characteristics. This facilitates better global coordination and a more balanced workload of factories compared to shallow networks and heuristics. 2) We propose an LLM-informed reward model that exploits the prior knowledge of LLM to generate multifaceted proxy rewards for each action. This allows timely and informative feedback during execution, enabling more precise credit assignment under sparse reward compared to manual scalar rewards for scheduling. 3) We evaluate *LaRL* on 167 DFJSP instances with six scales. The extensive experimental results demonstrate that *LaRL* significantly outperforms state-of-the-art metaheuristics, heuristics, and RL algorithms, and shows promising generalization to instances that are much larger than those used in training.

## 2 RELATED WORK

**Scheduling algorithms for DFJSP.** Exist scheduling algorithms for DFJSP include metaheuristics, heuristics, and RL-based algorithms. Among them, metaheuristics, such as the memetic algorithm (Zhu et al., 2024), search for high-quality schedules by balancing global exploration and local exploitation. Over the last few years, various heuristics have been designed for DFJSP. For example, random search (Zabinsky et al., 2009) explores the solution space by uniformly sampling feasible solutions. Iterated greedy (Zhao et al., 2025) improves initial solutions through iterative destruction and reconstruction mechanisms. Dispatching rules (Huang et al., 2024b) prioritize operations or machines based on predefined rules. However, these methods often rely on handcrafted rules and lack adaptability in dynamic environments. With the development of RL, researchers have shifted to RL-based algorithms. Lei et al. (Lei et al., 2024) proposed a heuristic selection framework to choose among predefined heuristics based on the current state. Wang et al. (Wang et al., 2025b) introduced an end-to-end policy learning approach that directly maps raw environment features to scheduling actions. However, these RL-based algorithms still face challenges in achieving global coordination across factories and in credit assignment under sparse rewards. This paper addresses the challenges via an end-to-end algorithm integrating global assignment and enhanced reward modeling.

**LLMs-assisted decision-making.** LLMs have recently emerged as powerful tools for complex decision-making tasks because of their advanced reasoning abilities and rich prior knowledge (Achiam et al., 2023). In particular, some recent studies have applied LLMs as planners to make decisions through APIs or predefined skills (Wang et al., 2024; Zhang et al., 2023a). For example, Liu et al. (Liu et al., 2023) encode problem descriptions into a formal prompt to achieve long-horizon planning. Valmeekam et al. (Valmeekam et al., 2023) highlight the potential of LLMs in structured planning tasks. Beyond their planning capabilities, LLMs possess remarkable code generation ability that facilitates the automation of function design and decision (Jiang et al., 2024). Recent works have shown that LLMs can generate, debug, and optimize code snippets, significantly accelerating development cycles (Zhong et al., 2024). Inspired by these promising works, we pro-

pose leveraging the dual strengths of LLMs in planning and code generation to assist RL in solving DFJSP, aiming to improve effectiveness and learning efficiency in dynamic environments.

## 3 PROBLEM DEFINITION

In DFJSP with new job arrivals, there are $n$ successively arriving jobs to be processed on $l$ distributed factories, to optimize a scheduling objective (e.g., *makespan* in this paper). Each factory $F_f(0 \leq f < l)$ is equipped with $M_f$ machines, and each job $J_i(1 \leq i \leq n)$ has a sequence of $k$ operations. Each operation $O_{ij}(1 \leq j \leq k)$ is assigned to one of its candidate machines, with its processing time ($p_{ij}$) depending on the selected machine. Solving DFJSP involves three key tasks: 1) assigning each new job to a specific factory; 2) selecting a machine for each operation within the assigned factory; and 3) determining the processing sequence of operations on each machine. After the above tasks, we can derive a *schedule*, i.e., the start times ($S_{ij}$) of each operation and their corresponding machine assignments, such that the makespan $C_{max} = max_{ij}\{C_{ij} = S_{ij} + p_{ij}\}$ is minimized subject to all constraints. In line with prior work (Lei et al., 2024), some assumptions are adopted as follows: 1) The transfer time of jobs is neglected. 2) All factories and machines are available at time zero. 3) Each machine can only process one operation at a time. 4) Each operation must be processed without interruption. 5) Each operation cannot be started until its previous operation is completed. In summary, DFJSP aims to assign new jobs to distributed factories and schedule their operations on machines to minimize makespan. The dynamic and sequential nature of DFJSP makes it well-suited for RL frameworks, which motivates our formalization as a Markov decision process.

## 4 PROPOSED ALGORITHM

This section presents the proposed LLM-assisted RL algorithm, LaRL, for DFJSP with new job arrivals. We begin with an overview of LaRL, followed by its three components: LLM-driven factory assignment, multi-agent collaborative scheduling, and LLM-informed reward model.

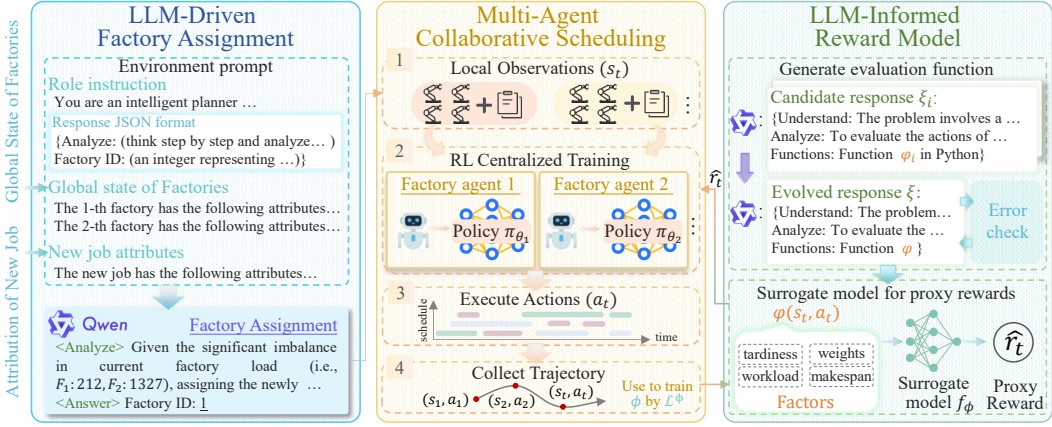

Figure 1: Overview of LaRL, which consists of three main components: (1) **LLM-driven factory assignment** assigns newly arrived jobs to factories by reasoning over the structured environment prompt consisting of the global factories state and new jobs attributions. (2) **Multi-agent collaborative scheduling** selects operation-machine pairs $a_t$ within each factory based on the local observations $s_t$, and the trajectories are collected for training. (3) **LLM-informed reward model** generates proxy rewards using a learned surrogate model $f_\phi$, which leverages the evaluation function $\phi$ generated by LLM to decompose action contributions across multiple dimensions.

### 4.1 OVERVIEW

As shown in Figure 1, LaRL consists of three components. First, LaRL uses LLM to guide job-factory assignment by encoding real-time states (e.g., load, availability, job features) into structured

prompts, and infers the most suitable factory via semantic reasoning. Second, factory-specific multi-agent groups select operation-machine pairs based on local observations in a decentralized manner. Third, to address sparse rewards, we introduce an LLM-informed reward model. Specifically, the LLM defines a semantically interpretable evaluation function mapping state-action pairs to multidimensional factors. We then train a surrogate model to estimate proxy rewards. This enables more accurate attribution of action contributions, facilitating stable and efficient policy learning. Our approach leverages the contextual understanding of LLMs to connect linguistic knowledge with symbolic decision-making, enhancing both effectiveness and generalization in DFJSP environments.

## 4.2 LLM-Driven Factory Assignment

To address the challenge of global coordination in DFJSP with new job arrivals, we propose an LLM-driven factory assignment mechanism that leverages the contextual reasoning capabilities and domain knowledge embedded in LLM. This mechanism determines the most suitable factory for each new job based on the global state, which is achieved by the following two steps 1) and 2).

1) *Environmental prompt*: The environment prompt serves to encode task-specific knowledge and contextual cues into a structured format, enabling the LLM to perform interpretable reasoning over the system state for decision-making. To this end, we construct the prompt $P$, defined as $P = Concat(R, G, A)$, where $R$ denotes the role instruction assigning a role to LLM and describing the problem profile and objectives, $G = G_1, \ldots, G_l$ presents the global states of all $l$ factories (e.g., workload, machine availability, and estimated delay ratio), $A$ encodes all attributions of the new job (e.g., weights, due date, and expected time).

2) *LLM-based factory assignment*: Given the constructed prompt, the LLM can evaluate the relationship between the new job and each factory based on the encoded information, and generate factory assignments, where both the selected factory ID and its analysis are returned in JSON format. The assignment result is then passed to the downstream multi-agent collaborative scheduling. Our design not only enables more effective global coordination but also improves interpretability compared to shallow networks and heuristics.

## 4.3 Multi-Agent Collaborative Scheduling

To enable efficient scheduling within each factory, we formulate the factory-level scheduling problem as a multi-agent decision process. Each agent selects an operation-machine pair based on local observations using a designed policy, while coordination is ensured through centralized training. The local observation, policy, and actions are set as follows.

1) *Local observation*: At decision step $t$, the agent of each factory $F_f$ receives $O_t^f \in \mathbb{R}^{m \times d}$, where $m$ is the number of machines, $d$ is the feature dimension. Each row in $O_t^f$ represents a ready operation and consists of the processing time matrix $P_t$, the operation feature matrix $F_t^o$, and the machine feature matrix $F_t^m$, i.e., $O_t^f = [P_t, ||F_t^o||F_t^m]$. To address the varying number of ready operations, the input size is fixed to $m$. When more than $m$ operations are ready, the top $m$ candidates with the earliest due dates are selected. Otherwise, we apply zero-padding with a binary mask to filter out invalid rows. The details of the two feature matrices are provided in the ***Appendix B***.

2) *Policy*: The policy $\pi$ outputs a probability distribution over all actions to determine which action is selected for execution. Considering the scheduling process is highly frequent and requires rapid response, we adopt a graph attention network (GAT) (Veličković et al., 2018)-based policy architecture rather than LLM, where all agents share the same policy architecture but maintain their own learnable parameters to adapt to the factory-specific constraints. For each agent, given the local observation $O_t^f$, the policy computes the action distribution in three stages. First, GAT is used to encode the raw features of each machine $M_m$ with its compatible ready operations $\mathcal{N}_t(M_m)$ into a $v$-dimensional embedding. By inputting $P_t$ and $F_t^m$, GAT computes importance weights for compatible operations and aggregates their features to update the machine embeddings as Eq. (1):

$$e_t^m = \sigma(\alpha_{mm} W_M F_t^m + \sum_{O_{ij} \in \mathcal{N}_t(M_m)} \alpha_{ijm} W_O F_t^o) \tag{1}$$

where $\sigma$ is an activation function, $W_M$ and $W_O$ are learnable matrices, $\alpha_{mm}$ and $\alpha_{ijm}$ are the attention coefficients representing the importance of machines to themselves and compatible operations.

Second, we use a multi-layer perceptron (MLP) to map the raw feature vectors of the ready operations into $v$-dimensional embeddings, as shown in Eq (2):

$$e_t^o = MLP_{\omega_0}(ELU[MLP_{\omega_1}(F_t^o)|| \sum_{m \in \mathcal{M}_t(O_{ij})} (e_t^m)]) \qquad (2)$$

where $\mathcal{M}_t(O_{ij})$ is the candidate machines of $O_{ij}$, $\omega_0$ and $\omega_1$ are learnable parameters of the MLPs.

Finally, we derive the policy distribution over all feasible operation-machine pairs by concatenating the learned operation and machine embeddings $[\mathbf{e}_{ij}^o, \mathbf{e}_{ij}^c]$ and passing them through another MLP followed by a softmax layer in Eq. (3):

$$P(a_t^f, O_t^f) = softmax(MLP_{\omega_2}(e_t^o||e_t^m)) \qquad (3)$$

where the $MLP_{\omega_2}$ with learnable parameters $\omega_2$ consists of two hidden layers and Tanh activation.

3) *Action*: A valid operation-machine pair from the ready set corresponding to each factor $F_f$. Specifically, the action can be defined as $a_t^f = (O_{ij}, m_h)$, which indicates that the operation $O_{ij}$ from job $J_i$ is assigned to an idle machine $m_h \in M_f$. The action is selected from a masked probability distribution over the feasible operation-machine combinations, where infeasible actions (e.g., machines not in the candidate set) are masked out to ensure valid execution. Based on these decisions, we collect the joint trajectories $\tau = \{(S_t, a_t)\}_{t=1}^T$ for all factories, where $S_t = \{O_t^1, \ldots, O_t^l\}$ and $a_t = \{a_t^0, \ldots a_t^l\}$. These trajectories are used for centralized training of the agent polices via Proximal Policy Optimization. More training details are provided in ***Appendix D***.

## 4.4 LLM-INFORMED REWARD MODEL

To enhance credit assignment in RL under sparse rewards, we propose an LLM-informed reward model that leverages the prior knowledge and reasoning capabilities of LLMs. This model addresses two key challenges: 1) how to effectively ask the LLM to produce helpful reward signals that are reliable and consistent with symbolic in RL and DFJSP, and 2) how to use these signals to better assign credit to actions taken at different time steps. To this end, the proposed LLM-informed reward model comprises two core components, i.e., generating evaluation functions and training a surrogate model for proxy rewards, as illustrated in Figure 1.

1) *generating evaluation functions*: Inspired by previous work (Qu et al., 2025), we adopt a two-stage generation process consisting of LLM-based generation and self-evolution phases. In the generation phase, we first construct a structured prompt by encoding the role instruction, problem description, global scheduling state, and agent action formats, detailed in ***Appendix A***. Then, the prompt is passed to LLM to produce $z$ candidate responses $\{\xi_i, \ldots, \xi_z\}$, each of which involves an executable code of an evaluation function $\varphi_i$. In the self-evolution phase, these functions are reorganized into the prompt. It guides the LLM to summarize a refined function $\varphi$ that integrates the strengths of candidates while reducing redundancy and inconsistency, as shown in Eq. (4).

$$\varphi = LLM(problem, role, \varphi_1, \ldots, \varphi_z) \qquad (4)$$

Furthermore, to ensure the executability of $\varphi$, we perform a preliminary error check by testing $\varphi$ on a random state-action pair. If any runtime errors occur, the corresponding error logs $err$ are appended to the prompt to guide the LLM in refining the function again, as detailed in Eq. (5).

$$\varphi = LLM(problem, role, \varphi, err) \qquad (5)$$

This two-stage process ensures that $\varphi$ not only captures semantically meaningful aspects of agent actions, but is also syntactically executable and aligned with the underlying scheduling objectives.

2) *Surrogate model for proxy reward*: Considering $\varphi$ is a symbolic and potentially non-differentiable function, build a surrogate model $f_\phi$ parameterized by $\phi$ based on the return decomposition (Efroni et al., 2021). This model can approximate the mapping from the observation-action pairs to scalar rewards. Specifically, the model estimates a proxy reward $\hat{r}_t$ from $\varphi(s, a)$ as $\hat{r}_t = f_\phi(\varphi(s, a))$. To align the proxy rewards with the episodic returns $R(\tau)$ collected from trajectories $\tau$, we train the surrogate model by minimizing the loss function in Eq. (6).

$$\phi^* = arg \min_\phi \mathbb{E}_{\tau \sim \pi}[(R(\tau) - \sum_{t=1}^T f_\phi(\varphi(s, a)))^2] \qquad (6)$$

This design effectively bridges the symbolic reasoning of LLM and numerical reward of RL, improving the learning efficiency of agents under sparse rewards, especially in large-scale scenarios.

## 4.5 THEORETICAL ANALYSIS

To further justify the effectiveness and generation of the proposed LaRL algorithm, we provide a theoretical analysis. Specifically, we analyze how well the learned surrogate model $f_\phi$ can approximate the true return $R(\tau)$ by leveraging the intermediate symbolic evaluation $\varphi(s, a)$. We define the approximation error and then provide an upper bound on this error.

**Theorem 1 (Reward Approximation Bound).**
*Assuming that $\exists f^* : \varphi(s, a) \mapsto r$, such that the true reward $r(s, a) = f^*(\varphi(s, a))$. Define the least-squares estimation error as $||r - \hat{r_\phi}||_{A_k}$, where $A_k = \sum_{i=1}^{k} \varphi(s_i, a_i)^\top \varphi(s_i, a_i) + \lambda I$. Then for any $\delta \in (0, 1)$, with probability at least $1 - \delta$, the estimation error satisfies the following concentration bound:*

$$||r - \hat{r_\phi}||_{A_k} \leq \sqrt{TD \log(1 + \frac{kT^2}{\lambda\delta})} + \sqrt{\lambda D} \tag{7}$$

where $T$ is the episode length, $D = dim(\varphi(s, a))$ is the factor dimension of the evaluation function.

**Theorem 2 (LLM-riven Factory Assignment Improves Global workload Balance).**
*Let $\mathbf{L}_T = [L_T^1, \ldots, L_T^l]$ denote the cumulative workload over $l$ factories after scheduling $n$ jobs. Suppose each job $J_i$ arrives from a stationary distribution and has a bounded processing time $p_t \in [0, p_{max}]$. If each job is assigned using an LLM-driven policy $\pi_{LLM}$ with bounded assignment error $\epsilon$, then the expected workload variance satisfies:*

$$\mathbb{E}[Var(\mathbf{L}_T)] \leq \frac{C}{T} + \epsilon^3 \cdot p_{max}^2 \tag{8}$$

where $C$ is a constant that represents the baseline workload variance caused by the job arrival distribution. The proofs of Theorems 1 and 2 are provided in ***Appendix C***.

## 5 EXPERIMENTS

### 5.1 EXPERIMENTAL SETUP

**1) Datasets.** To evaluate LaRL[1], we conducted experiments on benchmark-based and simulation-based datasets. The benchmark datasets are derived from Hurink (Hurink et al., 1994) and Brandimarte (Brandimarte, 1993) with two identical factories and varying jobs over 30, 50, 100, following the convention (Zhang et al., 2024). The simulation-based datasets include 1,000/2,000/5,000 sequentially arriving jobs, following the practice (Zhang et al., 2024). Details are in ***Appendix E***.

**2) Peer competitors.** We compare LaRL with six representative state-of-the-art algorithms from three categories. The first includes three popular heuristics: random search (RS) (Zabinsky et al., 2009), iterated greedy (PBIGA) (Zhao et al., 2025), and a dispatching rule (AR_SPT) (Huang et al., 2024b). The second is a representative metaheuristic, i.e., RMA (Zhu et al., 2024). The third is two state-of-the-art RL-based algorithms: PPOS (Lei et al., 2024) and P-G (Wang et al., 2025b).

**3) Parameter Settings.** The training settings follow (Li et al., 2024) with the batch size 128 and the initialized learning rate $1 \times 10^{-4}$ (decayed by 0.96/epoch). The policy employs a GAT with single-head attention with ELU activation, and an output embedding dimension of eight. The surrogate reward model is implemented as a three-layer MLP with ReLU activation and a hidden size of 256. We use the public LLM, *Qwen-max*, for reasoning in LaRL. More details are in ***Appendix E***.

**4) Evaluation Criteria.** We evaluate the performance using the average makespan (Mspan) over instances of each dataset. To assess load balance, we report the workload ratio (WR), defined as the ratio between the maximum and minimum total workloads across factories. Lower Mspan and WR closer to one indicate better performance.

### 5.2 RESULTS AND ANALYSIS

In this section, we evaluate LaRL against peer competitors on DFJSP instances with varying scales. The evaluation is based on the *average* makespan (Mspan) and workload ratio (WR) and statistical significance is assessed using the Wilcoxon rank-sum test ($p < 0.05$), where symbols '+', '=', and '−' indicate that LaRL performs significantly better, equivalent, or worse than the competitors.

---

[1]Code: https://anonymous.4open.science/r/LaRL-407B

### 5.2.1 PERFORMANCE ON INSTANCES WITH TWO FACTORIES

Table 1 presents the comparative results of LaRL and six representative scheduling algorithms on six datasets with varying scales. LaRL achieves the best WR on all datasets and outperforms all baselines in makespan on four of six instances, demonstrating its strength in global coordination and workload balancing. In terms of makespan, on small-scale instances (DFJSP-30/50/100), LaRL performs on par with or better than P-G, and consistently surpasses heuristic and metaheuristic baselines. On large-scale instances (DFJSP-1,000/2,000/5,000), it achieves the lowest makespan and WR across the board. In terms of WR, LaRL reduces WR to near 1.0 on all instances, with 1.16 on DFJSP-5,000, while others exceed 1.4, indicating superior workload balancing. All improvements are statistically significant under the Wilcoxon signed-rank test ($p < 0.05$), indicating that LaRL consistently outperforms existing algorithms in both makespan and workload balancing.

Table 1: Comparative study of different algorithms on instances with two factories.

| Algorithm | DFJSP-30 | | DFJSP-50 | | DFJSP-100 | | DFJSP-1,000 | | DFJSP-2,000 | | DFJSP-5,000 | |
|---|---|---|---|---|---|---|---|---|---|---|---|---|
| | Mspan | WR | Mspan | WR | Mspan | WR | Mspan | WR | Mspan | WR | Mspan | WR |
| RS | 2885.35(+) | 1.975(+) | 3536.59(+) | 2.17(+) | 6912.34(+) | 1.66(+) | 56241.32(+) | 1.85(+) | 127401.85(+) | 1.95(+) | 268174.60(+) | 2.07(+) |
| PBIGA | 2434.51(+) | 1.32(+) | 3506.24(+) | 1.13(=) | 6895.34(+) | 1.42(+) | 52889.34(+) | 1.40(+) | 119447.74(+) | 1.69(+) | 277748.52(+) | 1.62(+) |
| AR_SPT | 2696.47(+) | 1.55(+) | 3429.61(+) | 1.50(+) | 6599.55(+) | 1.43(+) | 53141.93(+) | 1.36(+) | 111066.90(+) | 1.31(+) | 267955.80(+) | 1.96(+) |
| RMA | 2422.35(+) | **1.03(=)** | 3482.51(+) | 1.31(+) | 6382.62(+) | 1.53(+) | 53317.53(+) | 1.48(+) | 117374.21(+) | 1.59(+) | 277512.01(+) | 1.69(+) |
| PPOS | 2262.50(+) | 1.79(+) | 3229.15(+) | 1.69(+) | 6367.15(+) | 1.55(+) | 52994.87(+) | 1.64(+) | 111302.20(+) | 1.53(+) | 267729.40(+) | 1.48(+) |
| P-G | **2096.24(+)** | 1.41(+) | **3176.34(=)** | 1.36(+) | 6266.21(+) | 1.36(=) | 52581.76(=) | 1.20(+) | 107297.36(+) | 1.65(+) | 266834.27(+) | 1.41(+) |
| LaRL (Ours) | 2141.00 | 1.15 | 3180.22(=) | **1.12** | **6254.50** | **1.31** | **52241.50** | **1.07** | **107374.00** | **1.03** | **266537.53** | **1.16** |

### 5.2.2 PERFORMANCE ON INSTANCES WITH MORE FACTORIES

To further evaluate the generalization of LaRL under different factory configurations, we evaluate LaRL on scenarios with three factories. Table 2 reports the average makespan and workload rate of all algorithms. Across all problem scales, LaRL consistently outperforms competitors in minimizing makespan, demonstrating its generalization ability when deployed in environments with more distributed factories. Notably, LaRL maintains balanced workload distribution among factories, as indicated by the reported workload rates, which remain close across factories. In contrast, peer competitors often suffer from workload skew, especially in large-scale instances (e.g., 2,000 and 5,000 jobs). These results confirm that the reasoning-based assignment and scheduling mechanism can effectively maintain both scheduling quality and workload balance in increasingly complex settings.

Table 2: Comparative study of different algorithms on instances with three factories.

| Scale ($f$-$m$-$n$) | (3-10-30) | | (3-10-50) | | (3-10-100) | | (3-10-1,000) | | (3-10-2,000) | | (3-10-5,000) | |
|---|---|---|---|---|---|---|---|---|---|---|---|---|
| Algorithm | Mspan | WR | Mspan | WR | Mspan | WR | Mspan | WR | Mspan | WR | Mspan | WR |
| RS | 2499.27(+) | 2.67(+) | 3036.62(+) | 2.19(+) | 6289.12(+) | 3.32(+) | 59253.71(+) | 2.69(+) | 119471.21(+) | 2.91(+) | 268174.60(+) | 3.71(+) |
| PBIGA | 2362.41(+) | 1.45(+) | 2911.34(+) | 1.28(=) | 6233.52(+) | 1.46(+) | 58756.86(+) | 1.54(+) | 120733.74(+) | 1.34(+) | 278450.33(+) | 3.27(+) |
| AR_SPT | 2233.16(+) | 2.15(+) | 2925.52(+) | 1.42(+) | 5994.70(+) | 1.24(+) | 56886.37(+) | 1.62(+) | 119281.73(+) | 1.47(+) | 269655.73(+) | 2.76(+) |
| RMA | 2491.17(+) | 1.58(+) | 2999.46(+) | 1.13(+) | 6187.05(+) | 1.53(+) | 57898.27(+) | 1.59(+) | 121241.17(+) | 1.39(+) | 283474.71(+) | 3.20(+) |
| PPOS | 2143.27(+) | 1.56(+) | 2950.00(+) | 1.38(+) | 6050.71(+) | 1.32(+) | 57083.15(+) | 1.47(+) | 110362.09(+) | 1.60(+) | 269972.53(+) | 2.09(+) |
| P-G | 2096.24(+) | 1.68(+) | 2927.71(=) | 1.29(+) | 5996.05(+) | 1.34(=) | 56483.62(+) | 1.37(+) | 110210.47(+) | 1.51(+) | 269547.63(+) | 1.83(+) |
| LaRL (Ours) | **2048.50** | **1.22** | **2841.86** | **1.04** | **5820.06** | **1.17** | **55205.66** | **1.16** | **109648.66** | **1.12** | **268870.06** | **1.23** |

### 5.2.3 ROBUSTNESS TO THE FREQUENCY OF JOB ARRIVALS

To evaluate the robustness of LaRL under different frequencies of job arrivals, we vary the utilization level from the default high-utilization setting (0.95) to a moderate level (0.85), which is commonly adopted in the scheduling literature (Zhang et al., 2023b). Experiments are conducted on instances with two factories, each equipped with 10 machines, and varying job sizes $1,000, 2,000, 5,000$. The comparison results are summarized in Table 3. We can observe that across all scales, LaRL consistently achieves lower makespan than peers, with particularly notable improvements in the large-scale setting with $5,000$ jobs. These findings confirm that LaRL is not only effective under high-pressure environments but also maintains superior when the frequency of job arrivals decreases.

Table 3: Comparative study of different algorithms in the scenario with a utilization level of 0.85.

| Method | DFJSP-30 | | DFJSP-50 | | DFJSP-100 | | DFJSP-1,000 | | DFJSP-2,000 | | DFJSP-5,000 | |
|---|---|---|---|---|---|---|---|---|---|---|---|---|
| | Mspan | WR | Mspan | WR | Mspan | WR | Mspan | WR | Mspan | WR | Mspan | WR |
| RS | 2215.73(+) | 1.69(+) | 3855.36(+) | 1.35(+) | 6062.15(+) | 1.81(+) | 60280.38(+) | 2.91(+) | 132015.47(+) | 3.26(+) | 320781.53(+) | 2.26(+) |
| PBIGA | 2090.34(+) | 1.49(+) | 3650.06(+) | 1.14(=) | 5801.75(+) | 1.63(+) | 61259.82(+) | 2.41(+) | 132154.22(+) | 2.16(+) | 325649.76(+) | 1.98(+) |
| AR_SPT | 2696.47(+) | 1.55(+) | 3732.71(+) | 1.30(+) | 5948.28(+) | 1.72(+) | 60745.58(+) | 1.96(+) | 129916.15(+) | 1.96(+) | 319984.66(+) | 1.87(+) |
| RMA | 2178.26(+) | 1.41(=) | 3415.29(+) | 1.21(+) | 5868.10(+) | 1.51(+) | 62046.62(+) | 2.33(+) | 132516.37(+) | 2.01(+) | 328975.61(+) | 2.09(+) |
| PPOS | 2078.31(+) | 1.51(+) | 3370.29(+) | 1.27(+) | 5810.53(+) | 1.35(+) | 59251.58(+) | 1.45(+) | 165368.73(+) | 1.59(+) | 310786.91(+) | 1.76(+) |
| P-G | **2006.73(-)** | **1.02(-)** | 3227.20(+) | **1.15(=)** | **5736.61(-)** | 1.31(+) | 59259.95(=) | 1.49(+) | 122259.23(+) | 1.76(+) | 308276.36(+) | 1.69(+) |
| LaRL (Ours) | 2026.56 | 1.09 | **3180.66(=)** | 1.17 | 5764.36 | **1.28** | **58986.26** | **1.18** | **120126.65** | **1.11** | **293905.67** | **1.21** |

### 5.2.4 ABLATION STUDY OF THE PROPOSED LLM-DRIVEN FACTORY ASSIGNMENT

To evaluate the contribution of the LLM-driven factory assignment, we replace it with two alternative strategies: (1) a heuristic-based method using the classic AR rule (Huang et al., 2024b), and (2) a learned neural network that maps global factory states and job attributes to assignment decisions (Lei et al., 2024). These two variants are referred to as LaRL_AR and LaRL_NN, respectively.

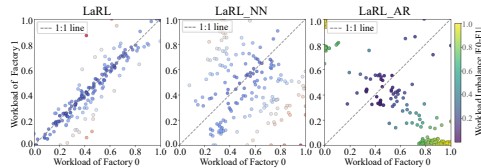

Figure 2: Comparison of workload between LaRL with two variants on two factories.

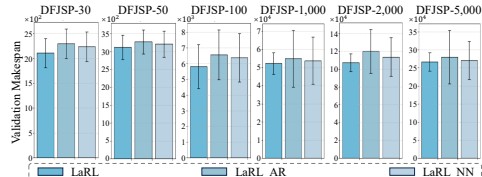

Figure 3: Comparison of makespan between LaRL with two variants on six datasets.

Figure 2 visualizes the factory-level load distribution using scatter plots. Each subfigure shows the normalized workload assigned to both factories on all instances, with the 1:1 diagonal line (gray dashed) indicating perfect workload balance. The color intensity reflects the absolute difference of workload ($|F0 - F1|$), where darker colors indicate smaller differences between factories. LaRL exhibits the most concentrated distribution along the diagonal line, reflecting highly superior balance. In contrast, LaRL-AR shows frequent skewed allocations due to static rules, while LaRL-NN shows moderate imbalance with less consistency. These results highlight the advantage of LLM-based assignment in leveraging global context and semantics to coordinate job allocation more effectively. Figure 3 shows that LaRL achieves consistently lower average makespan and variance across six datasets, indicating both superior performance and robustness. Notably, as the scale increases, the variance of LaRL remains significantly lower, highlighting its stability under complex settings. This suggests that the LLM can integrate diverse features and reason contextually, while rule-based or shallow models often rely on limited criteria, yielding instability under large-scale instances.

### 5.2.5 ABLATION STUDY OF THE PROPOSED LLM-INFORMED REWARD MODEL

To evaluate the contribution of our LLM-informed reward model, we conduct an ablation study by replacing it with a widely used handcrafted reward function based on makespan minimization (Lei et al., 2024), denoted as LaRL-HR. Both variants share the same policy architecture and training pipeline, differing only in the reward calculation.

Table 4 presents the comparative results between LaRL and its variant LaRL_HR. Across all six datasets with increasing problem scales, LaRL consistently achieves lower makespan and better or comparable WR on all instances. In particular, the relative advantage of LaRL is more evident in larger-scale settings. These results demonstrate that the multi-factor reward signals derived by LLM enable more effective credit assignment during training, leading to improved scheduling performance. Figure 4 illustrates the episodic return curves under two different shop utilization levels (0.95 and 0.85). In both settings, LaRL consistently converges faster and exhibits more stable learning with narrower shaded regions compared to LaRL_HR, indicating improved learning efficiency and robustness. Notably, under a higher utilization level (i.e., more frequent job arrivals), LaRL

Table 4: Comparison of Mspan and WR between LaRL and LaRL_HR on six datasets.

| Testing | LaRL | | LaRL_HR | |
|---|---|---|---|---|
| Datasets | Mspan | WR | Mspan | WR |
| DFJSP-30 | 2141.00 | 1.15 | 2198.72(+) | 1.27(+) |
| DFJSP-50 | 3180.22 | 1.12 | 3210.64(+) | 1.16(+) |
| DFJSP-100 | 6254.50 | 1.31 | 6379.46(+) | 1.29(=) |
| DFJSP-1,000 | 52241.50 | 1.07 | 52976.96(+) | 1.13(+) |
| DFJSP-2,000 | 107374.00 | 1.03 | 112951.21(+) | 1.09(+) |
| DFJSP-5,000 | 266537.53 | 1.16 | 269792.68(+) | 1.23(+) |

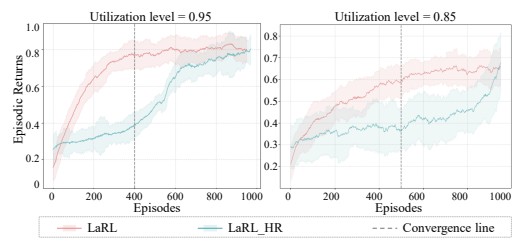

Figure 4: Episodic return curves of LaRL and LaRL_HR under two utilization levels, with smoothed returns (lines) and deviation (shade).

achieves faster convergence. This validates that our LLM-guided reward model not only enhances credit assignment under sparse feedback but also adapts well to varying scheduling intensities.

### 5.2.6 IMPACT OF LLM CHOICE

To evaluate the impact of different LLMs on the performance of LaRL, we compare LaRL using Qwen-max (default), DeepSeek-V3, and ChatGPT-3.5 on the benchmark-based datasets.

As shown in Table 5, although all variants benefit from LLM, stronger models such as ChatGPT-3.5 achieve lower average makespan and balanced workload. Qwen-max, adopted as the default because of its strong open-source accessibility and stable reasoning quality, achieves competitive performance across all settings. These results suggest that LaRL is robust to LLM choice and can further improve when equipped with more powerful models, highlighting its potential as a scalable framework for practical deployment.

Table 5: Comparison of LaRL using different LLMs on the benchmark-based datasets.

| Datasets | DFJSP-30 | | DFJSP-50 | | DFJSP-100 | |
|---|---|---|---|---|---|---|
| Algorithm | Mspan | WR | Mspan | WR | Mspan | WR |
| Deepseek-V3 | 2152.00 | 1.16 | 3190.88 | 1.12 | 6341.75 | 1.26 |
| ChatGPT-3.5 | 2092.60 | 1.05 | 3079.00 | 1.13 | 6217.67 | 1.21 |
| Qwen-max | 2141.00 | 1.15 | 3180.22 | 1.12 | 6254.50 | 1.31 |

### 5.2.7 ANALYSIS OF TIME COMPLEXITY

Let $n$ denote the number of jobs, $m$ the number of machines per factory, $l$ the number of factories, $d$ the average number of ready operations per machine, and $v$ the embedding dimension. Heuristics run in $\mathcal{O}(nlogn)$ to $\mathcal{O}(nm)$, while the metaheuristic RMA has a complexity of $\mathcal{O}(nmg)$ with $g$ as the population size. RL-based baselines typically involve per-step policy inference with complexity $\mathcal{O}(v^2)$ and $\mathcal{O}(lvn^2)$. LaRL introduces additional cost from the LLM-based factory assignment and GAT-based multi-agent scheduling, leading to an overall complexity of $\mathcal{O}(T_{LLM} + ldv^2)$. Despite this, LaRL remains highly efficient in practice, as the LLM is only invoked when new jobs arrive, and the scheduling decisions are made locally within each factory. More runtime cost comparisons are provided in **Appendix F.1**.

### 5.2.8 MORE EXPERIMENTS ARE IN THE APPENDIX

More experiments include evaluations on fewer machines (i.e., five machines per factory) (**Appendix F.2**), choice of GNN architectures(**Appendix F.3**), and interpretability of factory assignment (**Appendix F.4**). Additionally, practical applications of LaRL are discussed in **Appendix G**.

## 6 CONCLUSION

This paper addresses two core challenges: limited effective global coordination and the difficulty of credit assignment under sparse rewards. To this end, we propose an LLM-assisted RL algorithm, LaRL, which leverages the reasoning capability and domain knowledge of LLMs to guide factory assignment and construct multi-factor rewards. LaRL improves coordinated scheduling and efficient training, yielding better makespan and workload balance across scales.

ETHICS STATEMENT

All authors affirm that we have read and adhered to the ICLR Code of Ethics. Our study focuses on algorithmic development for distributed flexible job-shop scheduling using reinforcement learning and large language models. The datasets employed in our experiments are either publicly available benchmark datasets (Hurink and Brandimarte) or simulation-based synthetic instances, which do not involve human subjects, sensitive personal data, or identifiable information. No ethical concerns related to privacy, security, discrimination, or fairness are raised by our research. The proposed methods are intended to improve manufacturing efficiency in distributed scheduling environments and are not designed for harmful applications. All experiments comply with principles of research integrity, and no conflicts of interest or external sponsorships affect the reported results.

REPRODUCIBILITY STATEMENT

We have made extensive efforts to ensure the reproducibility of our work. All experimental setups, parameter configurations, and evaluation criteria are described in Section 5.1 and Appendix E. Theoretical assumptions and complete proofs of our main results are provided in Appendix C. Details of feature design, prompt construction, and training procedures are documented in Appendix A–D. To further support reproducibility, we release our full source code and experimental scripts via an anonymous repository (link provided in the abstract). These resources collectively ensure that our results can be independently verified and extended.

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

## A  LLM PROMPTS AND RESPONSE

We design our prompt following the chain-of-thought technique Wei et al. (2022). Below are the prompt templates and problem description in the LLM-driven factory assignment.

---

**Prompt Template**

**ROLE INSTRUCTION:**
You are an intelligent planner in a multi-factory scheduling system. Your task is to assign a newly arrived job to one of several factories, based on the job and factory attributes. You should first analyze the situation in natural language (briefly), and then choose the best factory. Your output must be **a single valid JSON object**, with exactly two keys:
1. 'Analysis': a brief explanation of your reasoning
2. 'Factory': the selected factory ID

Important rules:
- Your output must start directly with the curly brace
- Do NOT include any Markdown, code blocks, or extra text
- Do NOT write 'json', 'Answer:', or anything before or after the JSON
Example format:
```
{ Analysis:  Factory 1 has the most idle machines and shortest
delay, Factory:  1 }
```

**PROBLEM DESCRIPTION:**
You are tasked with making job-to-factory assignment decisions in a distributed production environment. The goal is to assign each newly arrived job to the most appropriate factory, balancing the workload, avoiding overload, and considering potential delay risks. You should make decisions based on the following attributes:

**NEW JOB ATTRIBUTIONS:**
The newly arrived job has the following attributes:
0: task_ID (int): The ID of the task to which this new arrival job belongs.
1: weight, due (int): the weight of the job is weight, e.g.,(0: normal, 1: urgency).
1: arrival_time (float): the time the job arrived in the system.
2: due_date (float): the time the job is due to be completed.
3: expected_time (float): the expected processing time of the job.
4: num_operations (int): the number of operations required to complete the job.

**FACTORY ATTRIBUTIONS:**
Each factory has the following attributes:
0: id (int): The ID of the factory.
1: number_machines (int): the number of machines in the factory.
2: average_utilization (float): the average utilization of the machines in the factory.
3: assigned_jobs (int): the number of jobs currently assigned to the factory.
4: earliest_start_time (list): the earliest available time of each machine in this factory.
5: idle_ratios (float): the current proportion of idle machines.
6: estimated_delay_ratio (float): the proportion of expected delayed jobs.

**Note:**

1. Your output must start directly with the curly brace.

2. Do NOT include any Markdown, code blocks, or extra text.

3. Do NOT write 'json', 'Answer:', or anything before or after the JSON.

---

Below are the prompt templates and problem description in the LLM-informed reward model.

**Prompt Template**

**ROLE INSTRUCTION:**
You are good at understanding job shop scheduling problems and writing Python code. You should fully understand the provided task and describe the exact observation and action form at the current decision point. Then, based on your understanding and the goal of the problem, analyze potential positive and negative behaviours or statuses that can be reflected in the observation and action. Finally, write an evaluation function that returns factors evaluating the current status from different aspects.
**Note:**

1. Do not use information you are not given!

2. Focus on the most relevant evaluation factors and use information in observation as little as possible.

3. The code should be as generic, complete and not contain omissions!

4. Avoid dividing by zero!

5. The input variable 'states' is a 3D tensor with shape (batch_size, num_factories × m, 20), representing the concatenated local observations from all factories; the input variable 'actions' is a 2D tensor with shape (batch_size, num_factories), where each entry indicates the index of the selected operation-machine pair or 0 if no action is taken; m means the number of machines in each factory.

6. Please return a list of several evaluation factor arrays, each in the form of (batch_size, 1).

7. Avoid all kinds of index out-of-bound errors! Always check index validity before indexing into the observation or action tensor.

Please think step by step and must adhere to the following JSON format (just replace the () with your answer):

```
{
    Understand: (your thoughts about the task),
    Analyze: (analyze behavior/status in observation/action),
    Functions: (define the Python function)
}
```

**SELF-PROMPTING:** You have generated several evaluation functions. Please summarize them and generate a new evaluation function that incorporates all the evaluation factors. If there are other important evaluation factors, please include them as well.

**Problem Information of DFJSP**

**PROBLEM DESCRIPTION:**
This is a distributed flexible job shop scheduling (DFJSP) with new job arrivals, involving the following components:
0. **Problem Overview**
- Multiple jobs arrive dynamically and need to be scheduled across multiple factories.
- Each factory contains a unique set of machines. Each job must be fully processed within a single factory.
- Each operation in a job can be assigned to one of several candidate machines, each with different processing times.
1. **Scheduling Goal**:
- Learn a global scheduling strategy that minimizes the makespan across all jobs in all tasks.
2. **Task Definition**:

- A task contains a set of jobs that must be scheduled jointly to optimize an objective.

3. **Job Structure**:

- Each job consists of a sequence of dependent operations.
- Each job is associated with:
- **Arrival time**: when the job becomes available for processing.
- **Due date**: the deadline by which the job should ideally be completed.
- **Weight**: a scalar indicating the urgency of the job; higher weights imply higher priority.
- These attributes affect the reward function and scheduling decisions.

4. **Operation Characteristics**:

- Each operation is only ready after all its preceding operations are completed.
- Each operation can be processed by a subset of machines in a factory, each with a distinct processing time.

5. **Machine Attributes**:

- Each machine can process only one operation at a time.
- Processing times vary per operation.

6. **Factory Constraints**:

- Each factory has a set of machines and can independently process entire jobs.
- All operations of a job must be executed within the same factory.

7. **Agent Policy Design**:

To minimize global makespan, we propose using a multi-agent reinforcement learning framework, where each factory is equipped with its own scheduling agents. Each agent represents a machine in a factory and is responsible for scheduling operations within its local factory scope, while indirectly cooperating to minimize makespan (time to complete all jobs). These agents may coordinate or operate independently to make real-time decisions for operation assignments within their local factory scope, while indirectly cooperating to minimize makespan (time to complete all jobs).

**STATE FORM:**

The state is concatenated from the observations of all agents at each decision point. The observations of each agent $S \in \mathbb{R}^{m \times 20}$ is a matrix representation at each decision point, where $m$ denotes the maximum number of ready operations in the system at that time step. Each row $s[i, :]$ corresponds to a specific ready operation, and encodes both operation-level and machine-level information relevant to decision making.

- $S[i, : m + 6]$: Each row represents one ready operation. A ready operation is an unprocessed operation whose all predecessor operations have been completed. If $s[i, :]$ is a zero vector (i.e., all elements are 0), it indicates a **padding row** used to maintain a fixed input shape when the number of ready operations is less than $m$.

- $S[:, 0 : m]$: A machine-operation processing time matrix. Each entry $S[i, j]$ denotes the processing time of the $i$-th ready operation on machine $j$. A value of 0 indicates that the operation cannot be processed on that machine. Each row represents one ready operation.

- $S[:, m : m + 6]$: Operation-level features for each ready operation:

   1. Job ID to which the operation belongs.
   2. Job weight (priority) for the operation.
   3. Average processing time of this operation across all candidate machines.
   4. Number of machines that can process this operation.
   5. Number of remaining unprocessed operations in the same job.
   6. Estimated tardiness of the job.

- $S[:, m + 6 :]$: Machine and factory-level features relevant to the operation:

   1. Current utilization of the candidate machine.
   2. Current workload of each machine.
   3. Earliest available time of each machine.

4. Factory ID to which the corresponding job has been assigned.

This unified representation captures both compatibility between ready operations and machines, as well as global scheduling context, enabling reinforcement learning agents to make informed and efficient scheduling decisions under dynamic job arrivals and machine disturbances.

**ACTION FORM:**
The dimensions of the multi-agent action are equal to the number of factories in the system. Each dimension corresponds to a specific factory and represents the operation-machine assignment selected by the agent within that factory at the current decision point. Specifically, the value in each dimension is an integer between 0 and 101, where a non-zero value indicates the index of the selected operation-machine pair to be processed, and 0 means that the agent chooses not to process any operations at that decision point.

## An example generated by LLM-informed reward model

**Understand:**
The task is a distributed flexible job shop scheduling problem with dynamic job arrivals, aiming to minimize the global makespan. Observations include machine-operation compatibility, job attributes, and factory constraints. Actions assign operations to machines within factories. Evaluation should consider workload balance, machine utilization, prioritization of high-weight jobs, tardiness minimization, and padding row ratios to assess system efficiency.

**Summary:**
Initial evaluation functions analyzed factors like workload variance across factories, average weighted tardiness, machine utilization, and padding row ratios. These metrics reflect balanced resource allocation, adherence to job urgency, operational efficiency, and underutilization of ready operations. A comprehensive function should integrate these aspects while avoiding redundancy and ensuring numerical stability.

**Function:**

```python
def evaluation_func(states, actions):
    batch_size, total_rows, features = states.shape
    num_factories = actions.shape[1]
    m = total_rows // num_factories
    workload_variance = torch.zeros((batch_size, 1), dtype=torch.float32)
    avg_weighted_tardiness = torch.zeros((batch_size, 1), dtype=torch.float32)
    machine_utilization = torch.zeros((batch_size, 1), dtype=torch.float32)
    padding_ratio = torch.zeros((batch_size, 1), dtype=torch.float32)
    priority_alignment = torch.zeros((batch_size, 1), dtype=torch.float32)
    for b in range(batch_size):
        factory_workloads = []
        total_padding = 0
        total_operations = 0
        weighted_tardiness_sum = 0
        total_weights = 0
        priority_sum = 0
        for f in range(num_factories):
            start_idx = f * m
            end_idx = (f + 1) * m
            factory_state = states[b, start_idx:end_idx, :]
            padding_mask = torch.all(factory_state == 0, dim=1)
            total_padding += torch.sum(padding_mask).item()
            total_operations += m
            if factory_state.shape[0] > 0:
                machine_workload = torch.max(factory_state[:, m+6+2], dim=0)[0]
                factory_workloads.append(machine_workload)
            weights = factory_state[:, m+1]
            tardiness = factory_state[:, m+5]
            valid_mask = weights > 0
```

```
        weighted_tardiness_sum += torch.sum(weights[valid_mask] * tardiness[
            valid_mask]).item()
        total_weights += torch.sum(weights[valid_mask]).item()
        priority_sum += torch.sum(weights[valid_mask]).item()
    if factory_workloads:
        factory_workloads = torch.stack(factory_workloads)
        workload_variance[b] = torch.var(factory_workloads)
    if total_weights > 0:
        avg_weighted_tardiness[b] = weighted_tardiness_sum / total_weights
        priority_alignment[b] = priority_sum / total_weights
    if total_operations > 0:
        padding_ratio[b] = total_padding / total_operations
        machine_utilization[b] = 1 - padding_ratio[b]
return [workload_variance, avg_weighted_tardiness, machine_utilization, padding_ratio
    , priority_alignment]
```

# B  FEATURE MATRICES

At each step $t$, the operation feature matrix $F_y^o$ and machine featur matrix $F_t^m$ are defined as follows:

The operation features matrix $F_y^o$: For each ready operation, the feature vector has six elements:

1. job ID: the ID number of the ready operation to which the job belongs.
2. weight: the weight of the ready operation to which the job belongs.
3. processing time: the average processing time of the operation on its candidate machine.
4. number of candidate machines: the number of candidate machines for the ready operation.
5. number of unscheduled operation: the number of remaining unprocessed operations of the job to which the operation belongs.
6. tardiness: expected tardiness probability of the job to which the ready operation belongs.

The machine feature matrix $F_t^m$: For each machine, the feature vector has three elements:

1. utilization: the ratio of the busy time of the machine to the total production time.
2. workload: the total processing time of all allocated operations to the machine.
3. available time: the time when the machine to complete the last operation assigned to it.
4. flag: the ID of the factory to which this machine belongs.

# C  PROOF

We follow a standard regularized regression generalization analysis using tools from concentration inequalities Boucheron et al. (2003) and properties of least-squares estimators Stock (1987).

**Notations.** Let $\varphi(s, a) \in \mathbb{R}^D$ be the symbolic evaluation vector derived from the LLM-informed function, $r(s, a)$ be the true reward for state-action pair $(s, a)$. $f^* : R \to \mathbb{R}$ means the ground-truth reward mapping over $\varphi(s, a)$, i.e., $r(s, a) = f^*(\varphi(s, a))$. $\hat{r}_\phi = \phi^\top \varphi(s, a)$ is the predicted reward and $\phi \in \mathbb{R}^D$ is the parameters of the reward model.

**Assumptions.** We assume the true reward satisfies $r(s, a) = f^*(\varphi(s, a)) = \phi^{*\top}\varphi(s, a)$, i.e., the reward is a linear function of the evaluation factor vector $\varphi(s, a)$.

*Proof of Theorem 1:* The proof can be divided into three main steps. In the first step, we collect a dataset of $k$ samples $\{(\varphi_i, r_i)\}_{i=1}^k$, where $\varphi_i := \varphi(s_i, a_i)$, and train $\hat{\varphi}$ via $\hat{\varphi} = arg\min_\phi \sum_{i=1}^k (r_i - \phi\top\varphi_i)^2 + \lambda||\phi||^2$. Then the solution can be obtained as shown in Eq. (9).

$$\hat{\phi} = A_K^{-1}(\sum_{i=1}^k \varphi_i r_i), \quad A_k = \sum_{i=1}^k \varphi_i \varphi_i\top + \lambda\mathcal{I} \tag{9}$$

where $A_k \in \mathbb{R}^{D \times D}$ is the regularized design matrix and $\lambda > 0$ is regularization coefficient.

In the second step, we can obtain the error vector over collected samples as $\epsilon_i := r_i - \hat{r}_\phi(s_i, a_i) = \phi^{*\top}\varphi_i - \hat{\phi}\top\varphi_i$, and then $\epsilon = (\phi^* - \hat{\phi})^\top \varphi_i$. Therefore, the squared weighted norm of the error is

shown in Eq.(10).

$$||r - \hat{r}_\phi||^2_{A_k} = (\phi^* - \hat{\phi})^\top A_k (\phi^* - \hat{\phi}) \tag{10}$$

In the third step, let $||\varphi s, a|| \leq T$ and $||\phi*|| \leq 1$ according to the bound for regularized linear regression under bounded norm assumption, and then with probability at least $1 - \delta$, we can obtain $||\phi^* - \hat{\phi}||_{A_k} \leq \sqrt{Dlog(1 + \frac{kT^2}{\lambda\delta})} + \sqrt{\lambda D}$. Based on this, we can yield the bound, as shown in Eq. (11).

$$||r - \hat{r}_\phi||_{A_k} \leq ||\phi^* - \hat{\phi}||_{A_k} \cdot ||\varphi|| \leq \left( \sqrt{D \log \left(1 + \frac{kT^2}{\lambda\delta}\right)} + \sqrt{\lambda D} \right) \cdot T \tag{11}$$

To make the bound consistent with the units of accumulated reward over episodes of length $T$, we apply a scaling factor to Eq. (11) and conclude $||r - \hat{r_\phi}||_{A_k} \leq \sqrt{TD \log(1 + \frac{kT^2}{\lambda\delta})} + \sqrt{\lambda D}$. This completes the proof.

**Notations.** Let the system have $n$ factories and the load vector at time $t$ be $L_t = [L_t^1, \ldots, L_t^n]$, where each component represents the total remaining processing time at factory $F_i$. At each time $t$, a job $J_t$ is drawn from distribution $\mathcal{D}_J$ with $p(J_t) \in [0, p_{max}$. Let $\pi^*(J_t)$ assign the job to the factory $F^*$ with the lowest current load: $F^* = arg \min_{i \in [n]} L_t^i$. Let $\pi_{LLM}(J_t)$ be the assignment produced by the LLM via structured prompt reasoning, and the error indicator as $\delta_t = \mathbb{I}[\pi_{LLM}(J_t)] \neq \pi^*(J_t)]$.

**Assumptions.** Assume that the misassignment rate is bounded in expectation, i.e., $\mathbb{E}[\delta_t] \leq \epsilon$. This captures the cases where the LLM, while not always optimal, makes errors with controlled frequency under the job distribution.

*Proof of Theorem 2:* The proof can be divided into two main steps. In the first step, let the factory assignment at time $t$ be $f_t = \pi_{LLM}(J_t)$, then the workload can be updated as shown in Eq.(12):

$$L_{t+1}^{f_t} = L_t^{f_t} + p_t, L_{t+1}^i = L_t^i \quad \forall i \neq f_i \tag{12}$$

In the second step, we calculate the mean workload as $\bar{L}_t = \frac{1}{n} \sum_{i=1}^n L_t^i$ and define the variance as $Var(L_t) = \frac{1}{n} \sum_{i=1}^n (L_t^i - \bar{L}_t)^2$. Now, we consider two cases:

1. If $\pi_{LLM}(J_t) = \pi^*(J_t)$, then $f_t = f^*$, and load variance decreases or stays constant (as the most underloaded factory receives the new job);

2. If LLM chooses the wrong factory $f_t \neq f^*$, then the most loaded factory may be chosen, increasing variance.

Based on the above cases, we analyze the expected change in variance as:

$$\mathbb{E}[Var(L_{t+1})] = \mathbb{E}[Var(L_t)] + \Delta_t \tag{13}$$

where $\Delta_t \leq \epsilon \cdot p_{max}^2 - \frac{1}{T} \cdot \mathbb{E}_{\pi^*}[Var(L_t) - Var(L_{t+!})]$. Over $T$ time steps, the summing variance increments can be obtained as Eq.(14).

$$\mathbb{E}[Var(L_{t+1})] \leq Var(L_0) + T \cdot \epsilon \cdot p_{max}^2 - \sum_{t=1}^T \frac{c}{t} \tag{14}$$

By the standard estimate of the harmonic series, we have $\sum_{t=1}^T \frac{1}{t} = \log T + \gamma + o(1)$, hence $\sum_{t=1}^T \frac{1}{t} \sim \log T$. Based on this, we approximate $\mathbb{E}[Var(\mathbf{L}_T)] \leq \frac{C}{T} + \epsilon^3 \cdot p_{max}^2$. This completes the proof.

# D  TRAINING DETAILS

We adopt Proximal Policy Optimization (PPO) (Schulman et al., 2017) as the policy learning algorithm. The overall objective combines a clipped surrogate policy loss $\mathcal{L}_p(t)$, a value function loss $\mathcal{L}_{value}(t)$, and an entropy regularization term $\mathcal{H}(\pi_\theta(\cdot|s_t))$, as defined in Eq. (15).

$$\mathcal{L} = \mathbb{E}_t \left[ \mathcal{L}_p(t) + c_v \, \mathcal{L}_{value}(t) - c_e \, \mathcal{H}(\pi_\theta(\cdot|s_t)) \right], \tag{15}$$

where $c_v$ and $c_e$ are the value loss and entropy coefficients, respectively. $\mathcal{L}_{\mathrm{p}}(t) = \mathbb{E}_t \left[ \min \left( r_t(\theta) \, \hat{A}_t, \, \mathrm{clip}(r_t(\theta), 1 - \epsilon, 1 + \epsilon) \, \hat{A}_t \right) \right]$. Here, $r_t(\theta) = \frac{\pi_\theta(a_t|s_t)}{\pi_{\theta_{old}}(a_t|s_t)}$ is the probability ratio between the current and previous policies, and $\hat{A}_t$ denotes the advantage estimate computed using Generalized Advantage Estimation (GAE) to reduce variance and stabilize training. $\mathcal{L}_{\mathrm{value}}(t)$ measures the mean squared error between predicted state values and target returns. Based on the PPO objective defined above, the overall training process for LaRL is summarized in Algorithm 1.

---

**Algorithm 1:** Framework of the training process

---

1   $\pi_{\theta_1}, \ldots, \pi_{\theta_l} \leftarrow$ Initialize the agent policies for all factories;
2   $V_\psi \leftarrow$ Initialize the centralized critic network with parameters $\psi$;
3   $f_\varphi \leftarrow$ Initialize the surrogate model with learnable parameters $\varphi$;
4   $B \leftarrow$ Initialize a replay buffer for storing experience data of agents;
5   **for** $i \leftarrow 1$ **to** *the maximum training episodes $T$* **do**
6      **for** $t \leftarrow 1$ **to** *the maximum steps completing all jobs* **do**
7         $a_t \leftarrow$ Decide a joint action by the agent policies based on the current state $s_t$;
8         $s_{t+1} \leftarrow$ Execute $a_t$ and transfer to new state;
9         $r_t \leftarrow$ Calculate the proxy reward by the surrogate model $f_\varphi$;
10        Store the trajectory $\langle s_t, a_t, r_t, s_{t+1} \rangle$ in $B$;
11      **end**
12      **if** *the number of trajectories in $B \geq$ preset threshold* **then**
13        Estimate value targets and advantages using $V_\psi$;
14        Update the agent policies via policy gradient using eq. (15);
15        Update the critic network $V_\psi$ by minimizing value loss;
16        Update the surrogate model $f_\varphi$;
17      **end**
18   **end**
19   **return** the optimal agent policies.

---

The training process begins by initializing the agent policies for all factories, the centralized critic, and the surrogate model, together with a replay buffer for storing interaction data (lines 1-4). During each episode, agents first select joint actions according to their current policies (line 7). Then, they interact with the environment to obtain the next state and the proxy reward from the surrogate model (lines 8-9). After that, the resulting trajectories are stored in the replay buffer (line 10). When the buffer reaches the preset threshold, the agent policies and surrogate model are updated based on the stored trajectories (lines 13-16). By repeating these steps, the optimal agent policies can be obtained.

## E   MORE DETAILS OF EXPERIMENT STEP

### (1) Dataset Details

**Benchmark-based dataset.** Due to the absence of standard benchmarks specifically tailored for the DFJSP with new job arrivals, we follow the established convention in the scheduling community Yan et al. (2024) to adjust classical static benchmarks for our setting. In particular, we extend the well-known Hurink Hurink et al. (1994) and Brandimarte Brandimarte (1993) benchmarks by introducing stochastic job arrival processes and configuring multiple factories. Concretely, we simulate dynamic arrivals by assuming jobs arrive according to a Poisson process, and jobs are dispatched to factories with identical machine configurations. We construct three datasets, i.e., DFJSP-30, DFJSP-50, and DFJSP-100, based on these benchmarks, where the number in the dataset name denotes the number of dynamically arriving jobs.

**Simulation-based datasets.** To assess the scalability and generalization of our approach in large-scale and multi-task environments, we construct three simulation-based datasets: DFJSP-1,000, DFJSP-2,000, and DFJSP-5,000. There are 30 different instances with different seeds. In each instance, all jobs arrive sequentially following a Poisson distribution Snyder & Miller (2012). Each job consists of 1 to 10 operations, each operation can be processed by 1 to 10 candidate machines, and processing times are randomly sampled from 1 to 99. The due date is determined as 1.5 times the total processing time added to the arrival time Zhang et al. (2023b). Following standard practices Lei et al. (2023), we control the job arrival frequency and machine load via a utilization level parameter, which is set to 0.75, 0.85, or 0.95 across different scenarios.

Table 6: All main hyperparameter settings used in our LaRL.

| Component | Value / Setting |
|---|---|
| *Policy Training* | |
| Batch size | 128 |
| Learning rate (policy) | $1 \times 10^{-4}$ |
| Learning rate decay | 0.96 per epoch |
| GAT output dimension | 8 |
| Optimizer | Adam ($\beta_1$=0.9, $\beta_2$=0.99) |
| *Reward Model* | |
| Hidden layer | 3-layer MLP |
| Hidden size | 256 |
| Activation function | ReLU |
| Reward model learning rate | $5 \times 10^{-4}$ |
| *LLM Parameters* | |
| Model | Qwen-max |
| LLM temperature (assignment) | 0.0 |
| LLM temperature (reward generation) | 1.0 |
| LLM temperature (self-evolution phase) | 0.3 |

**(2) More details of parameter settings**

To ensure stable and consistent decision-making in job-factory assignment, we set the LLM temperature to 0.0, preventing randomness in the output and promoting deterministic behavior across similar prompts. In contrast, the LLM-based reward function generation leverages controlled randomness to improve exploration. Specifically, we adopt a higher temperature of 1.0 in the initial generation phase to encourage diverse evaluation factor proposals. During the self-evolution phase, the temperature is reduced to 0.3 to guide the model toward convergence while retaining limited variability. These configurations are in line with prior work on symbolic reasoning with LLMs Qu et al. (2025). All detailed settings in LaRL are summarized in Table 6. All experiments were conducted on a workstation equipped with an NVIDIA GeForce RTX 3090 GPU. The LLM API used for both factory assignment and reward generation is Qwen-max, accessed via the official API. More details are available at: https://anonymous.4open.science/r/LaRL-407B.

## F MORE EXPERIMENTS

### F.1 EVALUATION ON TIME COST

To fairly evaluate computational efficiency, we report both the full runtime, including LLM communication overhead (w/LLM), and the pure scheduling time, excluding API latency (w/o LLM). As shown in Table 7, LaRL achieves a competitive runtime compared with other RL-based algorithms in the w/o LLM setting. Although its training time is slightly higher than PPOS because of an additional MLP-based surrogate reward model, it remains comparable to GNN-based algorithms such as P-G. Regarding evaluation time, LaRL (w/o LLM) is lower cost than P-G and close to PPOS, demonstrating its efficiency in

Table 7: Average training and evaluation time cost (in seconds) for each algorithm on the instances with 100 jobs. 0 indicates the algorithms do not require training.

| Method | Training Time | Evaluation Time |
|---|---|---|
| RS | 0 | 0.29 |
| PBIGA | 0 | 77.97 |
| AR_SPT | 0 | 0.24 |
| RMA | 0 | 76.39 |
| PPOS | 21.76 | 13.75 |
| P-G | 40.21 | 27.41 |
| **LaRL (w/o LLM)** | **39.83** | **11.83** |
| **LaRL (w/LLM)** | **269.95** | **299.02** |

inference and policy execution. The ad-
ditional time cost observed in the w/LLM configuration stems from remote LLM calls during factory
assignment. This overhead is largely external to the algorithmic design and can be substantially re-
duced in practical deployment via local hosting or caching. Notably, this cost enables globally
coordinated assignments and interpretable reward modeling, which conventional RL methods lack.
Therefore, the reported results indicate that LaRL achieves a reasonable trade-off between schedul-
ing performance and computational cost.

## F.2  EFFECT OF VARYING THE NUMBER OF MACHINES

Table 8 reports the comparative results
on large-scale instances with the con-
figuration of five machines per factory.
We compare LaRL with three repre-
sentative algorithms, including AR_SPT
(heuristic), RMA (metaheuristic), and
P-G (RL-based), all of which demon-
strate outstanding performance among
their categories. Even under limited
machine resources, LaRL consistently
outperforms or matches the baselines
across all instances in both makespan
and workload ratio. Notably, the advan-

Table 8: Comparative results under the configuration of
two factories and five machines per factory.

| No. of jobs | 1,000 | | 2,000 | | 5,000 | |
|---|---|---|---|---|---|---|
| Algorithm | Mspan | WR | Mspan | WR | Mspan | WR |
| AR_SPT | 57294.13 | 1.67 | 116706.43 | 1.40 | 292208.52 | 1.78 |
| RMA | 57374.25 | 1.58 | 116279.55 | 1.97 | 302389.01 | 2.03 |
| P-G | 57140.22 | 1.29 | 114192.40 | 1.24 | 292882.45 | 1.71 |
| LaRL(Ours) | 57010.48 | 1.12 | 113258.88 | 1.18 | 289522.03 | 1.22 |

tages of LaRL become more evident as the problem scale increases, demonstrating its superior scal-
ability in resource-constrained environments. The results also highlight the effectiveness of LaRL
in maintaining workload balance under tighter scheduling capacities.

## F.3  IMPACT OF GNN CHOICE

Figure 5 presents the comparison between
LaRL (GAT) and its two variants using GCN
and GIN, where the left Y-axis (log scale)
shows the average makespan; the right Y-axis
shows the relative gap (%) to LaRL(GAT).
Across all scales, LaRL-GCN consistently
underperforms GAT, confirming that simple
message passing struggles to capture com-
plex operation-machine relations. LaRL-GIN
achieves marginal gains over GAT on DFJSP-
100 and DFJSP-2000, but falls behind on the
remaining instances, especially as problem size
increases. These results highlight that the at-
tention mechanism of GAT is more suitable
for distributed scheduling tasks, as it dynam-
ically weighs operation-machine compatibility

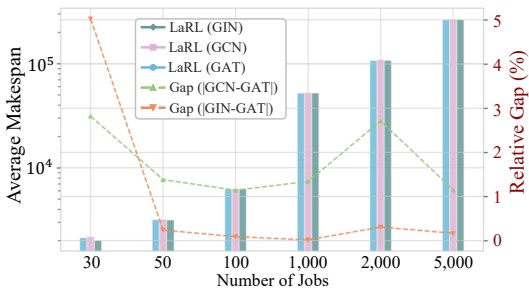

Figure 5: Comparison of LaRL variants with three
different GNN backbones (i.e., GCN, GIN, GAT)
on instances with varying scale.

and supports selective information aggregation under large-scale environments.

## F.4  INTERPRETABILITY ANALYSIS

To better understand the behavior of the proposed LLM-driven factory assignment module, we per-
form a qualitative and quantitative analysis of the reasoning generated by the LLM during new job
arrivals. We record the LLM-generated reasoning text along with the selected factory IDs. As shown
in Table 9, the LLM provides interpretable factory assignment decisions by explicitly reasoning over
key factors such as idle ratio, utilization, job urgency, and estimated delays. This highlights its ability
to transparently capture system dynamics and support human-understandable scheduling rationales,
validating the effectiveness of our prompt-based design in enhancing decision interpretability.

Table 9: Representative reasoning results from the LLM-based factory assignment.

| Job ID | Reasoning Summary | Factory |
|---|---|---|
| J1 | Factory 1 has the highest idle ratio (1.0) and no assigned jobs, making it the most available for processing the new job without delay risks. | F1 |
| J66 | Factory 0 has a low average utilization, a high idle ratio, and fewer assigned jobs, making it better suited to handle the new job without risking delays. Factory 1, although having some idle machines, already has more assigned jobs and shows signs of potential future load. | F0 |
| J100 | Factory 0 has a higher idle ratio (0.8) compared to Factory 1 (0.7), indicating more available capacity. Although both factories have similar utilization, Factory 0 also has a lower estimated delay ratio (2.17 vs. 1.43 in Factory 1 is worse). However, the weight of the job is normal and not urgent, so assigning it to the less loaded factory with better availability makes sense for balanced workload distribution. | F0 |

## G DISCUSSION

The integration of LLM into the scheduling pipeline, as demonstrated in LaRL, offers significant performance improvements in makespan and workload balance. Although the use of LLMs introduces additional inference overhead, LaRL remains practical for real-world manufacturing due to its decision frequency and deployment flexibility. In many production settings, factory assignment decisions are made at discrete intervals (e.g., upon new job arrivals), rather than at high frequency, making the time cost of LLM reasoning acceptable in practice. More importantly, the semantic reasoning ability provided by LLM enables globally balanced job allocation and interpretable reward shaping, which are critical for practical large-scale, dynamic environments. In addition, as LLMs continue to evolve toward more lightweight and efficient variants, the proposed LaRL framework offers a scalable foundation for intelligent scheduling systems in industrial applications.

## H THE USE OF LARGE LANGUAGE MODELS

In preparing this manuscript, we used LLMs, specifically ChatGPT, as a writing assistant. The LLM was employed to enhance the clarity, conciseness, and grammatical accuracy of the text, refine phrasing, and ensure a consistent academic style across sections. It was also used to rephrase technical descriptions for better readability and to polish the final presentation of this work. LLMs were not used for generating research ideas, designing algorithms, conducting experiments, or producing theoretical results. All contributions were developed exclusively by the authors.

