# OpenReview forum: "LLM-Assisted Reinforcement Learning for Distributed Scheduling"
_ICLR.cc/2026/Conference — ICLR 2026 Conference Withdrawn Submission_

### Official Review · Reviewer_LDFv · 2025-10-23

**Soundness:** 2
**Presentation:** 3
**Contribution:** 2
**Rating:** 2
**Confidence:** 4

**Summary:**

This paper proposes an LLM-assisted reinforcement learning framework for the distributed flexible job-shop scheduling problem (DFJSP). It uses LLM for factory assignment for newly arriving jobs, multi-agent apporach for machine scheduling within factories and   an LLM-informed reward model through evaluation functions and surrogate modeling.

**Strengths:**

The work addresses practically important problem of DFJS with dynamic job arrivals. The paper adapts existing FJSP benchmarks for evaluation, which could be useful for other work on DFJSP.

**Weaknesses:**

Even though the paper provides convincing experimental results, the LaRL framework lacks sufficient detail, and the use of LLMs is inadequately motivated. The authors cite LLMs' "domain knowledge" as justification (line 174), however, the paper does not demonstrate how this knowledge is incorporated beyond prompt-based problem descriptions.

The surrogate reward model is intended to solve the credit assignment problem using data generated by multiple RL agents. However, it is  unclear which reward function these individual RL agents follow during training. Additionally, the paper does not clarify how and why the RL agent of each factory interact with each other, or what specific objective guides their coordination. What is role of the reward model in RL centralized training?

No detail of baselines except for their name are given.  Please add few sentences explaining alteast PPOS and P-G.

**Questions:**

What is the added benefit of LLM for factory assignment? How about an evalution where LLM is replaced with a heuristic for factory selection problem?

Within the LLM-informed reward model: How exactly do the evaluation functions guide the surrogate model training? What is the actual input-output relationship?

---

### Official Review · Reviewer_5RT3 · 2025-10-29

**Soundness:** 2
**Presentation:** 2
**Contribution:** 2
**Rating:** 2
**Confidence:** 3

**Summary:**

The paper proposes LaRL, a RL method for the distributed FJSSP that integrates a LLM at two key points of the system. The authors argue that existing RL-based approaches struggle to balance workloads across factories because factory assignment decisions are based on heuristics or shallow networks, and that learning is inefficient due to sparse scalar rewards that provide little guidance during an episode. The paper therefore uses an LLM first to decide where newly arriving jobs should be processed: global factory states and job attributes are converted into a structured prompt, and the LLM chooses a factory together with a short rationale. Second, the LLM is used to generate code for a multi-factor evaluation function that decomposes state–action quality into interpretable dimensions. A surrogate model then learns from these factors to produce dense proxy rewards during training. This means the LLM influences both the global coordination of job assignment and the internal shaping of the learning signal.

Within each factory, operation scheduling is performed by a multi-agent RL architecture with graph attention networks, which handles the frequent local decisions independently of the LLM. Experiments on established benchmarks and large simulated instances show that the proposed method improves makespan and workload balance compared to heuristic, metaheuristic and modern RL baselines. Ablation results indicate that both LLM components, factory assignment and reward factorization, contribute meaningfully to performance.

**Strengths:**

- The paper addresses a relevant and challenging problem and presents a novel and coherent hybrid approach that leverages LLM reasoning for high-level decisions and RL for low-level scheduling, supported by extensive experiments and clear empirical improvements over strong baselines, which demonstrates the practical effectiveness and potential impact of the proposed idea.

- The paper introduces an interesting paradigm, using LLMs not just as decision-makers but as tools for reward shaping, which may inspire further research on combining LLM reasoning with RL in other domains.

- Despite some conceptual claims being under-argued, the empirical validation is thorough and consistently supports the main contribution: the proposed system performs better than strong baselines across diverse benchmarks. The method is described with enough detail to be implementable, and the ablations demonstrate that both LLM components contribute meaningfully to performance.

**Weaknesses:**

- The paper empirically shows that LaRL performs well, but does not establish a causal connection between the proposed LLM components and the observed performance gains. Therefore, the claimed mechanisms remain suggestive rather than demonstrated.

- The argumentation seems somewhat "marketing-heavy". Many claims are made without sufficient methodological justification. For example, the paper refers to "LLM reasoning" and "semantic global decision-making," but provides little systematic analysis of why and when the LLM actually makes better decisions.

- The description of how the reward model is created through LLM code generation remains conceptually vague. It is not not shown how stable or reproducible this process is, nor is it discussed.

- Although the authors include a theoretical bound on the surrogate reward model, the analysis remains largely superficial. The result is mathematically valid but generic. It essentially states that if the surrogate reward approximates the true reward sufficiently well, the resulting policy will also be near-optimal. However, the theory does not provide insight into why the LLM-generated reward factors should satisfy the required assumptions, nor does it explain how the bound relates to practical performance.

- Several architectural and design choices appear to be driven primarily by empirical observation (“this configuration works better in practice”), yet the paper presents them as theoretically motivated without offering deeper justification or comparison to alternatives. As a result, it is sometimes difficult to distinguish which components are conceptually essential and which are empirically tuned. Providing clearer rationale or more systematic ablations would strengthen the causal link between design decisions and performance.

- The paper does not discuss the computational or practical costs of repeatedly querying an LLM during training or inference. Since factory assignment depends on external LLM calls, factors such as latency, query failures, prompt sensitivity, or cost–performance trade-offs are ignored. This omission makes it difficult to assess whether the method is feasible in settings where decisions need to be taken under strict time constraints.

**Questions:**

- Can you provide quantitative or qualitative evidence that the LLM actually performs semantic reasoning during factory assignment?

- How stable is the reward factor generation across multiple LLM runs or prompt variations? Please provide more statistics or experiments across multiple seeds and prompt perturbations.

- How does the theoretical bound relate to the observed empirical behavior? Could you show how large the approximation error is in practice?

- Can you further elaborate which architectural choices were theoretically grounded and which were selected empirically?

- How well does the method generalize when realistic constraints are introduced, such as transfer times, communication latency, or partial observability? Do you have any insights about that?

- What is the computational cost of querying the LLM during training and inference, and is the approach viable in real-time scheduling scenarios?

- Is LaRL still superior if the surrogate reward is generated manually by domain experts (instead of by an LLM)?

- Can you clarify whether similar performance gains could be achieved through manually designed reward shaping (for example by domain experts)? In other words, what specific advantage does the LLM-based decomposition provide over conventional, hand-crafted reward functions?

---

> ### Comment · Reviewer_5RT3 · 2025-11-27
>
> Since the authors did not provide a rebuttal, I maintain my original evaluation.

---

### Official Review · Reviewer_3wzE · 2025-10-30

**Soundness:** 1
**Presentation:** 1
**Contribution:** 1
**Rating:** 2
**Confidence:** 4

**Summary:**

The paper deals with the distributed flexible job-shop scheduling problem (DFJSP) where a set of jobs is assigned to a set of factories to meet objectives such as minimising the makespan. It proposes an LLM-assisted RL algorithm, which includes three components: LLM-driven factory assignment, multi-agent collaborative scheduling, and LLM-informed reward model. However, the paper lacks of clarity in the description of LLMs' role in the problem.

**Strengths:**

The attempt of linking LLMs to DFJSP.

**Weaknesses:**

1. Based on the workflow, the LLM is used to pick factories and generate a code-based factor extractor that feeds a learned surrogate reward. However, we don’t see a clear coherent technical line about why these LLM-produced artifacts outperform simpler heuristics or human defined features.

2. In the multi-agent collaborative scheduling part, it is unclear to me how the LLM generated reward works. The execution time is a numerical value from each machine. Why LLM is needed for reward generation?

3. The theoretical analysis seems come from nowhere. It is hard to understand how these sections are put together.

**Questions:**

see weaknesses

---

### Official Review · Reviewer_17mq · 2025-10-30

**Soundness:** 2
**Presentation:** 2
**Contribution:** 2
**Rating:** 4
**Confidence:** 4

**Summary:**

This paper introduces LaRL, a large language model (LLM)-assisted reinforcement learning algorithm for distributed flexible job-shop scheduling (DFJSP). The method integrates an LLM-driven factory assignment mechanism for global coordination and an LLM-informed reward model to improve credit assignment under sparse rewards.

**Strengths:**

The paper tackles a challenging industrial scheduling problem with a creative integration of LLM reasoning and reinforcement learning. The framework is technically sound, combining theoretical analysis with empirical validation.

**Weaknesses:**

The mechanism of how the LLM reward generation concretely contributes to stable learning is not sufficiently justified beyond empirical outcomes. The dynamic and potentially unstable nature of evolving proxy rewards is not properly addressed. Moreover, the factory assignment component seems to assume a static or near-perfect assignment without feedback adaptation, leaving unclear how assignment errors propagate through multi-agent training. The analysis of reward signals and their stability could be strengthened with ablations or sensitivity tests.

**Questions:**

The model appears to assume that the factory assignment module performs well without retraining—does it adapt during RL updates, or is it fixed after initial generation?

Given that each factory agent operates independently, how does centralized training handle cases where global rewards provide misleading signals due to unbalanced factory performance?

What is the methodological rationale for the LLM reward model working effectively—beyond empirical observation, how does it theoretically ensure meaningful gradient signals for policy learning?

The role and necessity of the proxy reward model are not entirely clear; in which part of the training pipeline is it used, and how does it interact with the main reward from the environment?

As the reward function evolves through LLM self-improvement, how is reward signal stability ensured to prevent oscillations or inconsistent learning?

Section 5.2.5 compares different rewards, but does not explain whether distinct reward definitions lead to compatible reward curves. How is cross-reward comparison made meaningful across settings?

---

### Note · Authors · 2025-11-27

I have read and agree with the venue's withdrawal policy on behalf of myself and my co-authors.